# Short Communication: Quantifying and Correcting for Pre-Assay CO₂ Loss in Short-Term Carbon Mineralization Assays

Matthew A. Belanger[1,2], Carmella Vizza[1,2], G. Philip Robertson[1,3], and Sarah S. Roley[2]

[1]W.K. Kellogg Biological Station, Michigan State University, Hickory Corners, MI 49060, USA
[2]School of the Environment, Washington State University, Richland, WA 99354, USA
[3]Department of Plant, Soil, and Microbial Sciences, Michigan State University, East Lansing, MI 48824, USA

*Correspondence to*: Carmella Vizza (carmella.vizza@wsu.edu)

**Abstract.** The active fraction of soil organic carbon is an important component of soil health and often is quickly assessed as the pulse of $CO_2$ released by re-wetting dried soils in short-term (24−72 h) assays. However, soils can lose carbon (C) as they dry and if soil samples vary in moisture content at sampling, differential C loss during the pre-assay dry-down period may complicate the assay's interpretations. We examined the impact of pre-assay $CO_2$ loss in a long-cultivated agricultural soil at
initial moisture contents of 30, 50, and 70% water-filled pore space (WFPS). We found that 50 and 70% WFPS treatments lost more C during drying than did those in the 30% WFPS treatment, and that dry-down losses led to a 26−32% underestimate of their $CO_2$ pulses. We developed a soil-specific correction factor to account for these initial soil moisture effects. Future C mineralization studies may benefit from similar corrections.

## 1 Introduction

The short-term pulse of $CO_2$ following the re-wetting of dried soils (Robertson et al., 1999; Franzluebbers et al., 2000) has been widely used as an indicator of soil health because of its strong relationship to soil organic C, particulate organic C, microbial biomass C, and cumulative nitrogen and C mineralization over longer periods (e.g., 24 days; Franzluebbers et al., 2000). This method is derived from the "Birch effect," whereby re-wetted dry soils release a pulse of $CO_2$ resulting from increased microbial activity (Birch, 1958). Drought stress drives microbial communities to dormancy or death (Borken and
Matzner, 2009), and re-wetting stimulates C mineralization (Kim et al., 2012).

Several mechanisms could explain the Birch effect, reviewed by Jarvis et al., (2007), among others. Briefly, these include 1) drying and wetting destroys soil aggregates, thus making previously inaccessible organic substrates available (Denef et al., 2001; Homyak et al., 2018); 2) microbes killed after drying are decomposed upon re-wetting (Blazewicz et al., 2014; Blazewicz
et al., 2020); 3) microbes release solutes to avoid bursting in response to osmotic stress caused by re-wetting (Schimel et al., 2007); and 4) microbial populations and their activity rebound in response to re-wetting (e.g., Barnard et al., 2013). Recent studies suggest that both cellular and extracellular C are likely to contribute to the larger $CO_2$ pulse following re-wetting (Slessarev et al, 2020; Warren, 2020), implying that multiple mechanisms could be important.

Although the short-term pulse of $CO_2$ following the re-wetting of dry soils is a widely used method in soil health assessments (e.g., Culman et al., 2013; Ladoni et al., 2016; Morrow et al., 2016; Sprunger and Robertson, 2018), there may be potential bias introduced by assaying soils of different moisture contents at the time of sampling. Soils that differ in moisture will dry down at different rates, potentially losing different amounts of C prior to the start of the assay. If sufficiently large, differential pre-assay losses could complicate soil health comparisons across field treatments, landscape catenas, or different timepoints within the same soil.

Here we investigate the influence of initial soil moisture levels on pre-assay $CO_2$ release during drying for an Alfisol soil in the upper Midwest, USA. We test the hypothesis that moister soil will have higher pre-assay $CO_2$ loss because a longer dry-down period results in more time for such losses to occur.

## 2 Materials and methods

### 2.1 Site description

We collected soil using a shovel from the Ap horizon (0−20 cm) of an arable grass field at the W.K. Kellogg Biological Station (KBS) in Hickory Corners, MI (42°41'02" N, −85°37'34" W). KBS soils are mixed, mesic Typic Hapludalfs of co-mingled Kalamazoo and Oshtemo series (Crum and Collins, 1995) developed on glacial outwash with intermixed loess (Luehmann et al., 2016). Soil collected in September 2019 for this experiment was from the Kalamazoo series, which are well-drained fine-loams (43% sand, 38% silt, 19% clay) with ~2% total C (Grandy and Robertson, 2006) and a pH of 7.2 (Robertson et al., 1993). Average annual precipitation at KBS is 1005 mm, and mean annual temperature is 10.1°C (Robertson and Hamilton, 2015). The site was in various corn-soybean-wheat rotations for the 40 years prior to sampling and before that, corn-soybean-small grain rotations for at least 60 years.

### 2.2 Experimental design

To examine the influence of initial soil moisture on the pre-assay loss of $CO_2$ during dry-down, we pre-wet recently collected soil to three different initial water-filled pore space (WFPS) levels: 30, 50, and 70%. Then we measured gravimetric soil moisture (GSM) and $CO_2$ loss while soil was air-drying, after which we re-wet them and measured the 24-h $CO_2$ pulse by standard methods (Robertson et al., 1999; Franzluebbers et al., 2000).

### 2.3 Laboratory analyses

After collection, soil was sieved through a 4-mm mesh and mixed. We measured GSM and calculated the target volumetric water content (VWC, g $H_2O$ cm$^{-3}$ soil) for each treatment following Eq. 1 (Elliott et al., 1999):

$$VWC = WFPS/100 * (1- SBD/2.65) \tag{1}$$

where soil bulk density (SBD) is 1.5 g soil $cm^{-3}$, as previously assessed by Robertson (2016). Then we divided VWC by SBD to obtain a target GSM and thereby determined the amount of water to add to the field-moist soil (11% WFPS; GSM = 0.032 g $H_2O$ $g^{-1}$ dry soil). We then weighed 40 g of soil into each of 75 polyethylene cups. Each cup was randomly assigned to an initial WFPS treatment (30, 50, or 70%), for a total of 25 replicates per treatment. We added sufficient deionized water to each cup to achieve the target initial WFPS and stirred to evenly distribute water. After soil was wet and stirred in the cups, the contents of each cup were transferred to a labeled paper bag. The soil was spread evenly across the bottom of the bag, and the top portion of the bag was removed to increase air flow. Afterwards, the soil was immediately weighed and set on a laboratory bench to air-dry.

Immediately after wetting, as well as 1, 3, and 8 days later, we assessed GSM and $CO_2$ loss rates for five replicates per initial WFPS treatment. GSM, which was determined after drying the soil at 105°C for 24 h, stabilized at 1.5% in the air-dried soil (Fig. 1a), but did not reach zero even when soil was completely air-dry. Because soil in all initial WFPS treatments were air-dry by day 3, with $CO_2$ loss rates close to zero, we terminated GSM and $CO_2$ measurements after day 8.

$CO_2$ loss rates at each sampling interval were measured by placing 10 g of soil into a 235 mL mason jar fitted with a gas-sampling septum. Then we sampled 5 mL of headspace from each jar at 4 intervals (0, 0.5, 1, and 2 h), injected it into an evacuated 3 mL exetainer (Labco Limited, Lampeter, Wales, United Kingdom), and replaced the jar headspace with laboratory air. $CO_2$ samples were analyzed within 24 h using a LI-820 $CO_2$ Gas Analyzer (LI-COR Biosciences, Lincoln, NE, USA).

On day 15 we re-wet the remaining five replicates of air-dried soil from each initial WFPS treatment to 50% WFPS (Franzluebbers et al., 2000). We then assessed subsequent 24-h $CO_2$ pulses by sampling headspaces at 0, 2, 4, 8, and 24 h.

**2.4 Statistical analyses and correction factor**

$CO_2$ loss rates and pulses were calculated as the positive slope of the linear regression of $CO_2$ concentrations through time after accounting for headspace dilution, and then converted to a standardized rate using the ideal gas law. In 17 of 75 cases, we omitted one of the four data points within a jar, which were clear visual outliers. In two cases, we rejected jars with leaks. $CO_2$ loss rates during the dry-down period were analyzed with a two-way analysis of covariance (ANCOVA), where initial WFPS treatment and days elapsed since wetting (Day) were factors and GSM at the time of sampling was a covariate. Additionally, a one-way analysis of variance (ANOVA) was used to determine whether initial WFPS treatment significantly affected the 24-h $CO_2$ pulses upon re-wetting the air-dried soil.

We also calculated a correction factor to account for pre-assay $CO_2$ loss prior to the 24-h $CO_2$ pulse assay. To calculate the total amount of $CO_2$ loss during dry-down for each initial WFPS treatment, we calculated a best-fit exponential decay curve:

$$Y = \alpha^{\beta X} + \theta \tag{2}$$

where Y = daily $CO_2$-C loss and X = length of dry-down period, until soil was air-dry (i.e., immediately after wetting through day 3). Total C loss was equivalent to calculating the area under the curve, using bootstrapping to estimate error.

Because we used sacrificial sampling, we could not calculate standard deviation or standard error in the usual way. Instead, we used a bootstrapping approach in which we computed predicted values for $CO_2$ losses ($\hat{Y}_i$) and residuals ($e_i = Y_i - \hat{Y}_i$). All zeros for $CO_2$ losses were set to 1 for fitting the regression because an exponential decay curve can approach but never attain 0 and because 1 was lower than any value we observed. Then we created a bootstrap sampling of residuals specific to each dry-down interval (0, 1, or 3 days), sampled randomly from each interval with replacement, and added randomly sampled

residuals to predicted values ($Y_i^* = \hat{Y}_i + e_i^*$) for each dry-down interval (after Hesterberg, 2015). Residuals were bootstrapped 10,000 times to derive multiple estimates of coefficients for the exponential decay curve ($\alpha$, $\beta$, and $\theta$). We also integrated under the curve 10,000 times to get an error estimate (i.e., coefficient of variation) associated with the total amount of pre-assay $CO_2$ loss during dry-down.

Then we divided the total $CO_2$ loss by three days to obtain the daily rate used to calculate a correction factor following Eq. 3:

CF = (daily $CO_2$ loss during dry-down / 24-h $CO_2$ pulse after re-wetting) + 1        (3)

The correction factor for each treatment was then multiplied by each replicate's 24-h $CO_2$ pulse following re-wetting. Finally, we verified that the correction factors worked by conducting a one-way ANOVA to determine whether initial WFPS treatment still had an effect on the corrected pulses. For all analyses, we confirmed that assumptions of normality and homogeneity of

variance were not violated.

## 3 Results

As expected, soil in the 50 and 70% WFPS treatments took longer to dry than did soil in the 30% WFPS treatment (Fig 1a). A day after wetting, soil from the 30% WFPS treatment was completely air-dry, but soil in the 50 and 70% WFPS treatments had lost only 79% and 68% of their initial moisture, respectively. All soil was air-dry by three days after wetting. Pre-assay

$CO_2$ losses mirrored soil moisture loss, reaching zero for all WFPS treatments by day 3 (Fig 1b). Both GSM at the time of sampling and Day had effects on pre-assay $CO_2$ loss rates ($P < 0.0001$), but initial WFPS treatment did not ($P = 0.28$), probably because GSM captures more variation in soil moisture than WFPS treatment as the soil dries. However, there was an interaction between treatment and Day ($P = 0.0005$). Soil of even the lowest initial WFPS treatment lost C as $CO_2$ over three days of drying (26 µg $CO_2$-C $g^{-1}$ soil for 30% WFPS), but losses were disproportionately higher from wetter soil (62 and 71 µg $CO_2$-

C $g^{-1}$ soil for 50 and 70% WFPS, respectively).

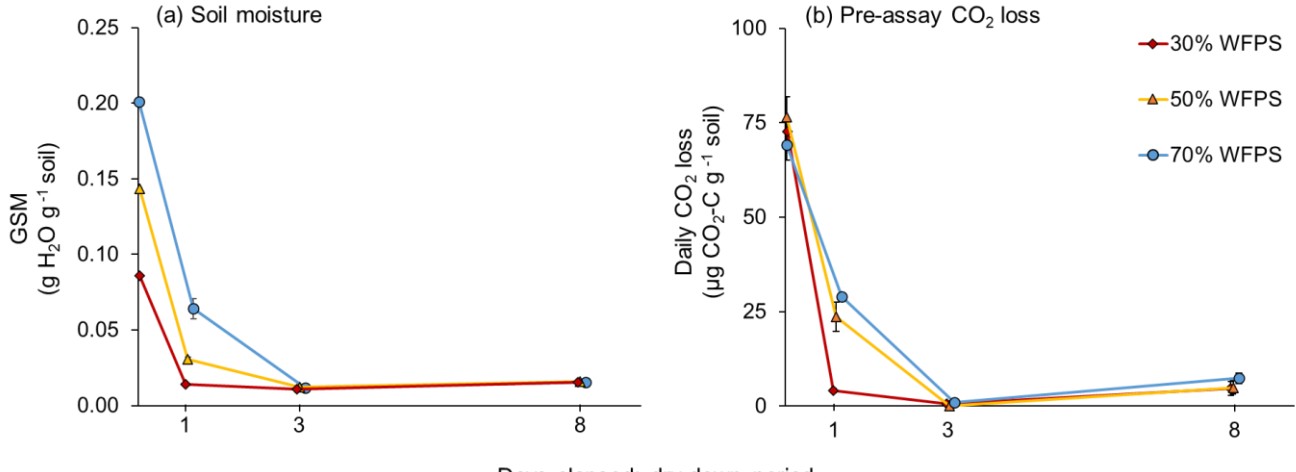

**Fig 1. (a) Gravimetric soil moisture (GSM) during air-drying and (b) daily CO₂ losses from each initial water-filled pore space (WFPS) treatment during the dry-down period. Error bars represent standard errors of the mean.**

Initial soil moisture (i.e., WFPS treatment) had a significant effect on 24-h $CO_2$ pulses after re-wetting air-dried soil ($P$ = 0.007; Fig 2). While final $CO_2$ pulses were lower for the 50 and 70% WFPS treatments relative to 30% WFPS (Fig. 2), the 50

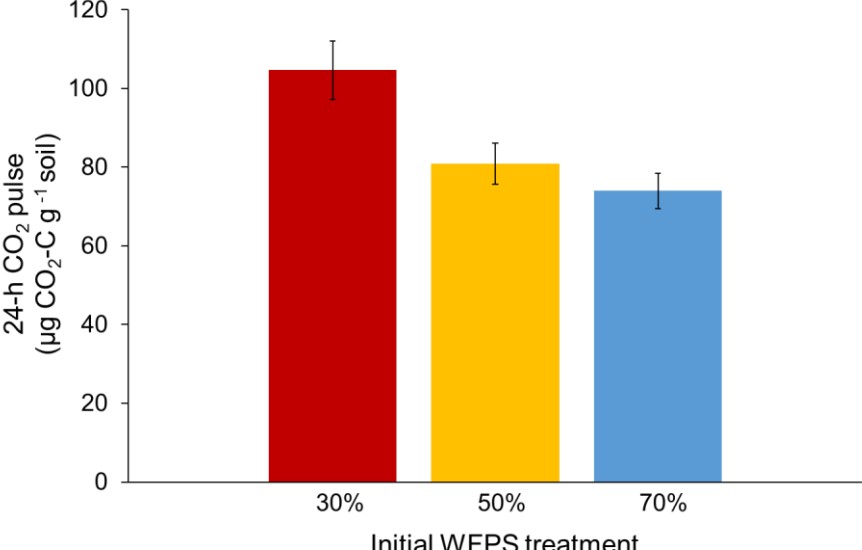

**Fig 2. 24-h CO₂ pulses after the re-wetting of air-dried soil for each initial water-filled pore space (WFPS) treatment. Error bars represent standard error of the mean.**

and 70% WFPS treatments also tended to have greater pre-assay $CO_2$ losses during three days of dry-down, which represented 77 and 95% of their 24-h $CO_2$ pulses, respectively. After accounting for these losses with correction factors, the 24-h $CO_2$ pulses were similar across initial WFPS treatments ($P = 0.28$; Fig. 3).

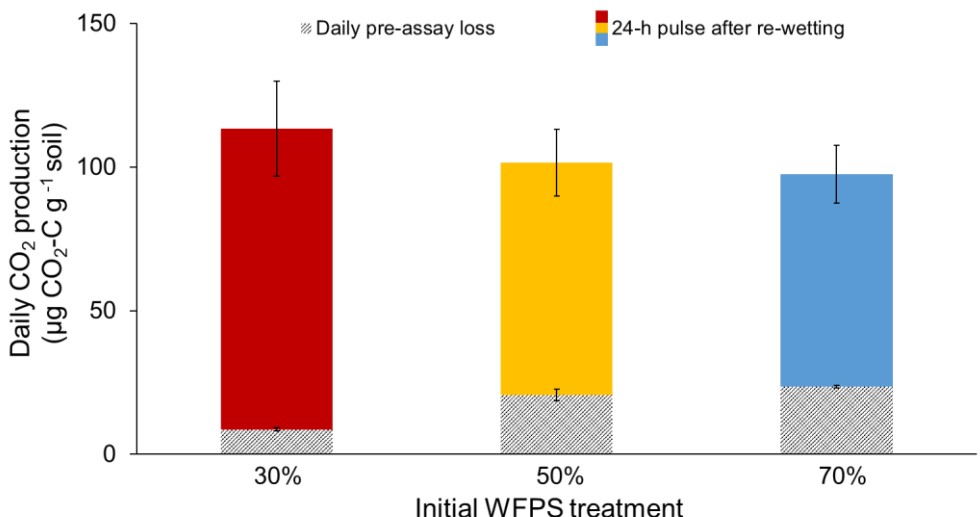

**Fig 3. Daily $CO_2$ production rates for each initial water-filled pore space (WFPS) treatment. Lined bars represent the average daily**
**rate of pre-assay $CO_2$ loss during a 3-day dry-down period and solid bars represent the 24-h $CO_2$ pulses after re-wetting the air-dried soil. Together both bars represent the 24-h $CO_2$ pulse corrected for pre-assay losses of $CO_2$ during dry-down. Error bars represent standard deviation, which was calculated based on bootstrap error propagation for the daily pre-assay losses.**

## 4 Discussion

Initial soil moisture levels significantly affected the results of the conventional 24-h $CO_2$ pulse assay, thus calling into question
its reliability as a soil health indicator (Fig 2). Wetter soil lost more C during dry-down, presumably because soil microbes remained active for a longer period of time. Without knowledge of pre-assay $CO_2$ losses, one might erroneously conclude that soil from the 30% WFPS treatment had about a 35% greater $CO_2$ pulse than the others (Fig 2), but this trend is instead due to higher pre-assay $CO_2$ losses during the dry-down period for wetter soil (Fig 3). It is striking that even short drying intervals (i.e., 1 versus 3 days) can affect soil health interpretations as deduced from the 24-h $CO_2$ pulse after re-wetting air-dried soil.
However, we were able to account for the pre-assay $CO_2$ losses for our soil with a correction factor that made 24-h $CO_2$ pulses approximately equivalent across all initial WFPS treatments. It is unlikely that any mechanism other than dry-down loss in wetter soil is responsible for the treatment differences we observed. First, soil samples were treated identically except for initial water content. Were potential mechanisms behind the Birch effect responsible, wetter soils should have released higher $CO_2$ pulses upon re-wetting due to greater microbial activity (e.g., Linn and Doran, 1984), more microbial biomass (e.g.,
Franzluebbers, 1999), or a greater release of osmolytes due to the increased risk of lysis. However, the 30% WFPS treatment had the largest $CO_2$ pulse upon re-wetting (Fig 2), so dry-down loss in wetter soils is the most plausible explanation (Fig. 1).

These trends suggest that efforts to characterize soil health via short-term $CO_2$ pulses following the re-wetting of dry soil should exercise caution if comparisons involve soils with a range of initial soil moistures. This includes soils compared across seasons; across drought, precipitation, or irrigation gradients; across landscape catenas; across crop, grazing, or forest management practices; and as well in cross-site comparisons and meta-analyses that include soils collected at different initial soil moistures.

A correction factor that accounts for pre-assay $CO_2$ losses may help to normalize such comparisons. In our soil, pre-assay $CO_2$ losses led to a C mineralization bias as high as 32%, for which we could confidently correct by applying a correction factor based on measured rates of pre-assay $CO_2$ loss (Eq. 3). However, we acknowledge that our use of a correction factor is not intended to imply that there is a fixed, available C pool, rather to demonstrate the way antecedent moisture conditions can affect soil health tests and to offer a potential solution. Other soils with moisture contents sufficient for microbes to oxidize available C during dry-down will require different correction factors as the C quality and microbial communities could change by soil type as well as within and across seasons (Groffman et al., 1996; Wuest, 2014). A soil-specific correction factor can be calculated by measuring $CO_2$ loss during dry-down on a subset of samples, as we described above (Eq. 3).

An alternate solution is to minimize the dry-down period such that little C is lost prior to the assay. Strategies to minimize pre-assay $CO_2$ loss include exposing soils to temperatures high enough to speed evaporation, but low enough to avoid sterilization (Jager, 1968) or otherwise artificially disrupt the microbial community (Evans and Wallenstein, 2012). This could be performed in a closed vented chamber such as a soil incubator. Alternatively, faster and more even drying might be achieved with a steady flow of air (i.e., a fan or vented system) over exposed soil samples.

Overall, our results demonstrate that using the 24-h $CO_2$ pulse following the re-wetting of a dried soil to evaluate soil health can be misleading for soils with different moisture contents at time of sampling. For such soils, rapid drying methods and/or a correction factor based on pre-assay $CO_2$ losses should be considered.

**Data availability**

Data is publicly available at Dryad: https://doi.org/10.5061/dryad.fj6q573rf.

## Author contributions

CV and GPR designed this study, MAB and CV performed the laboratory assays, MAB analyzed the $CO_2$ samples, CV conducted the statistical analyses, SSR and GPR obtained funding, MAB and CV wrote the paper with contributions from SSR and GPR.

## Competing interests

The authors declare that they have no conflict of interest.

## Acknowledgements

We acknowledge the Roley lab group at WSU and the Robertson lab group at KBS for their helpful comments, especially Sven Bohm, whose questions inspired us to undertake this study. We also thank Stuart Jones for his advice about bootstrapping. We thank two anonymous reviewers and editor Jocelyn Lavallee for their valuable feedback on how to improve this manuscript. Support for this research was provided by the National Science Foundation (NSF) Division of Environmental Biology DEB 1754212, by the NSF Long-term Ecological Research Program (DEB 1832042) at the W. K. Kellogg Biological Station (KBS), by the Great Lakes Bioenergy Research Center, U.S. Department of Energy Office of Science, Office of Biological and Environmental Research (Award DE-SC0018409), and by Michigan State University AgBioResearch.

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
