# Peer review of "Short Communication: Quantifying and Correcting for Pre-Assay CO₂ Loss in Short-Term Carbon Mineralization Assays"

_SOIL, 2020_

## Referee Comment (RC1) · Anonymous Referee #1 · 1 Nov 2020

This short communication looks at the methodological artifact of soil dry-down on measuring active soil carbon as CO2 burst. The topic would be of interest to readers of SOIL, although the manuscript may be even better fit for journal like SSSAJ since prevalence of this method in the US. I have some general concerns/suggestions and specific comments to help the authors improve their manuscript.

1. There is considerable research on soil moisture content and biological activity of soils, including dry-down and wet-up. This should be discussed more in Introduction and Discussion. There are several studies that show that the drier soils are in the field, they tend to release more CO2 upon re-wetting regardless of dry-down C loss. Thus,

there are biophysical mechanisms at play other than dry-down C loss. Figure 3 still shows more CO2-C from the 30

Borken, W., Davidson, E. A., Savage, K., Gaudinski, J., Trumbore, S. E. (2003). Drying and wetting effects on carbon dioxide release from organic horizons. Soil Science Society of America Journal, 67(6), 1888-1896.

Jarvis, P., Rey, A., Petsikos, C., Wingate, L., Rayment, M., Pereira, J., ... Manca, G. (2007). Drying and wetting of Mediterranean soils stimulates decomposition and carbon dioxide emission: the "Birch effect". Tree physiology, 27(7), 929-940.

Liang, L. L., Grantz, D. A., Jenerette, G. D. (2016). Multivariate regulation of soil CO2 and N2O pulse emissions from agricultural soils. Global change biology, 22(3), 1286-1298.

2. These findings need to be placed in context of the purpose and practicalities of CO2 Burst test. Drying is often needed, if not necessary, to stabilize microbial activity before analyzing for CO2 production. Whereas accounting for C loss during drying might not be feasible or possible for commercial and even research labs. This would be very onerous. How much more are we gaining by accounting for this C? This paper has not convinced me that we gain much. Despite finding differences between 30This paper would be much stronger if it included a comparison of two or more treatments, and showed that measuring CO2 Burst at different moisture contents obscured our ability to detect treatment differences. There are always experimental artifacts with incubation-based, laboratory measurements of soils. A few studies have shown that using more soil reduces variability. The most important thing is that we treat soils the same across time or space, and that the methodology is not creating confounding effects.

SPECIFIC COMMENTS:

L24. Replace 'the reintroduction of moisture' with 'rewetting'

L33. Delete 'different'

[Figure]

L38. How was it collected? Shovel, or soil probe? More details are needed here.

L43. So total MAP is 2305? This seems very high.

L49. What was the initial water content of the field soil when you collected it? Was it below 30

L85. This seems overly complicated. Why not use area-under-the-curve to calculate cumulative CO2?

L110. Why does respiration go back up at 8d? This is interesting and looks like there might be treatment effect?

L131. Why use standard deviation in this graph? Fig. 2 uses standard error. I suggest being consistent. Also, use same colors in Fig. 1 and 2 for consistency. Place letters to abbreviate significant differences among means in both Fig. 1 and 2.

---

## Author Comment (AC1) · 13 Nov 2020

1. There is considerable research on soil moisture content and biological activity of soils, including dry-down and wet-up. This should be discussed more in Introduction and Discussion.

**Response: For the sake of brevity in this "short communication", we did not review the many studies on soil moisture content and $CO_2$ pulses. However, we would be happy to include a more nuanced introduction and discussion if we are permitted to increase this manuscript's length. We also thank Reviewer #1 for pointing us towards these sources.**

There are several studies that show that the drier soils are in the field, they tend to release more CO2 upon re-wetting regardless of dry-down C loss. Thus, there are biophysical mechanisms at play other than dry-down C loss.

**Response: While we recognize that biophysical mechanisms influence $CO_2$ pulses after re-wetting dry soil (reviewed in Jarvis et al. 2007), they are not likely to differ between treatments because soil samples were treated identically except for initial water content. All soil samples were sieved and air-dried, thus destroying aggregates equally, and then re-wet to 50% WFPS for the 24-hr $CO_2$ pulse assay (Fig. 2). The only difference in moisture content between treatments occurred before air-drying. Additionally, these biophysical mechanisms do not account for the drier soil releasing the largest $CO_2$ pulse after re-wetting (Fig. 2). If anything, wetter soils should have released higher $CO_2$ pulses due to greater microbial activity (e.g., Linn and Doran 1984), more microbial biomass (e.g., Franzluebbers 1999), or a greater release of osmolytes due to the increased risk of lysis. However, the 30% WFPS treatment had the largest $CO_2$ pulse (Fig. 2), and dry-down loss in wetter soils is the most plausible explanation (Fig. 1).**

Figure 3 still shows more CO2-C from the 30.

**Response: Differences in total C loss among treatments were not significant (one-way ANOVA: P = 0.28). More importantly, the proportion of pre-assay loss was substantially smaller (Fig. 3).**

2. These findings need to be placed in context of the purpose and practicalities of CO2 Burst test. Drying is often needed, if not necessary, to stabilize microbial activity before analyzing for CO2 production. Whereas accounting for C loss during drying might not be feasible or possible for commercial and even research labs. This would be very onerous. How much more are we gaining by accounting for this C?

**Response: We agree that our findings are inconvenient for an assay that requires drying and that calculating a correction factor is an added complexity. As we point out in the discussion (lines 145-166), a correction factor is not needed if comparing soils of similar moisture contents at collection or perhaps if the dry-down period is fast. However, not correcting for soils at different moistures will likely lead to erroneous conclusions about carbon availability.**

This paper has not convinced me that we gain much. Despite finding differences between 30, this paper would be much stronger if it included a comparison of two or more treatments and showed that measuring CO2 Burst at different moisture contents obscured our ability to detect treatment differences.

**Response: We agree that pre-assay dry-down effects will likely differ not just by treatment but also by soil type and structure; this is why we discuss the need for calculating correction factors on a case-by-case basis (lines 151-155). Little would be gained by adding different treatments to this study because we fully expect correction factors to differ.**

There are always experimental artifacts with incubation-based, laboratory measurements of soils. A few studies have shown that using more soil reduces variability. The most important thing is that we treat soils the same across time or space, and that the methodology is not creating confounding effects.

**Response: We agree that it is important that methodology not create confounding effects, which is why we chose to conduct this study. Our results reveal that initial soil moisture can confound our interpretation of carbon availability from the $CO_2$ pulse assay. We want others to be aware of**

**this limitation, and so we provide a correction and a few alternatives for minimizing $CO_2$ losses during dry-down.**

SPECIFIC COMMENTS:

L24. Replace 'the reintroduction of moisture' with 'rewetting'

**Response: Agreed.**

L33. Delete 'different'

**Response: Agreed.**

L38. How was it collected? Shovel, or soil probe? More details are needed here.

**Response: We collected the soil using a shovel and will add this to the manuscript.**

L43. So total MAP is 2305? This seems very high.

**Response: No, snowfall has an average density of ~10% so 1300 mm represents only ~130 mm of precipitation. Nevertheless "average annual rainfall" should read "average annual precipitation" to include snowfall. Thank you for identifying this issue. We will change the sentence to make it clear that MAP is 1005 mm (not 2305 mm) and that the site receives 1.3 m of snowfall on average.**

L49. What was the initial water content of the field soil when you collected it? Was it below 30

**Response: Yes, the initial soil moisture was 11% WFPS (please see lines 56-57). We will move this to the collection section to clarify.**

L85. This seems overly complicated. Why not use area-under-the-curve to calculate cumulative CO2?

**Response: We did in fact calculate the area under the curve, using bootstrapping to define the curve to allow for error estimates. We will edit the text to clarify.**

L110. Why does respiration go back up at 8d? This is interesting and looks like there might be treatment effect?

**Response: The slight increase in $CO_2$ respiration at Day 8 may be due to more humid incubation conditions later in the experiment as evidenced by the slight increase in soil moisture (Fig. 1a).**

L131. Why use standard deviation in this graph? Fig. 2 uses standard error. I suggest being consistent. Also, use same colors in Fig. 1 and 2 for consistency. Place letters to abbreviate significant differences among means in both Fig. 1 and 2.

**Response: SD is more appropriate than SE for Fig 3, which reports error from the results of bootstrapping. Thank you for the suggestions about consistent colors and letters.**

**References**

**Borken, W., Davidson, E. A., Savage, K., Gaudinski, J., & Trumbore, S. E. (2003). Drying and wetting effects on carbon dioxide release from organic horizons. *Soil Science Society of America Journal*, *67*(6), 1888-1896. https://doi.org/10.2136/sssaj2003.1888**

Fierer, N., & Schimel, J. P. (2002). Effects of drying–rewetting frequency on soil carbon and nitrogen transformations. *Soil Biology and Biochemistry*, *34*(6), 777-787. https://doi.org/10.1016/S0038-0717(02)00007-X

Franzluebbers, A. J. (1999). Microbial activity in response to water-filled pore space of variably eroded southern Piedmont soils. *Applied Soil Ecology*, *11*(1), 91-101. https://doi.org/10.1016/S0929-1393(98)00128-0

Jarvis, P., Rey, A., Petsikos, C., Wingate, L., Rayment, M., Pereira, J., ... & Manca, G. (2007). Drying and wetting of Mediterranean soils stimulates decomposition and carbon dioxide emission: the "Birch effect". *Tree physiology*, *27*(7), 929-940. https://doi.org/10.1093/treephys/27.7.929

Liang, L. L., Grantz, D. A., & Jenerette, G. D. (2016). Multivariate regulation of soil $CO_2$ and $N_2O$ pulse emissions from agricultural soils. *Global change biology*, *22*(3), 1286-1298. https://doi.org/10.1111/gcb.13130

Linn, D. M., & Doran, J. W. (1984). Effect of water-filled pore space on carbon dioxide and nitrous oxide production in tilled and nontilled soils. *Soil Science Society of America Journal*, *48*(6), 1267-1272. https://doi.org/10.2136/sssaj1984.03615995004800060013x

---

## Referee Comment (RC2) · Anonymous Referee #2 · 14 Nov 2020

The manuscript by Belanger et al. describes a soil incubation experiment designed to quantify the effect of antecedent soil moisture on the amount of CO2 released from soil after drying and rewetting. The authors find that wetter soils emit more CO2 during dry-down and less CO2 after wet-up. In the soils studied here the additional C respired during drying approximately equaled the CO2 that was not respired after wet-up. The authors conclude that the C respired represents a fixed pool of available C, and hence soil tests that include respiration measurements from wetting of dried soil can be corrected to account for C lost during drying.

The basic finding of this paper seems well supported by the data: antecedent moisture

history affects the release of CO2 from dry soil following wetting. This observation seems generally consistent with previous studies: for instance, both the duration of drying (Miller et al. 2005; Meisner et al. 2015) and the severity of drying (Meisner et al. 2017) influence respiration after wetting. In this case, soil moisture during the wet period was varied and found to affect the respiration rate after wetting. The novelty of this short note is that it raises this point specifically in the context of soil health testing.

The specific interpretation advanced in this study–that respiration prior to drying affects the post-wetting respiration pulse specifically by reducing C availability–is only indirectly supported by the data and might need more thought. This interpretation seems to rest on an assumption that there is a fixed pool of available C at sampling, and that any losses of C between sampling and drying/rewetting reduce the size of this pool–resulting in a proportionately smaller pulse. Strictly speaking this is assumption is true of the bulk organic C pool, but it may not apply to the small fraction of that bulk pool that is actually available at any given moment (e.g. the soluble C pool). The apparent balancing of C fluxes observed in this experiment (Fig 3) does seem consistent with the idea of a fixed available C pool–but several factors could make things more complicated:

(1) Depolymerization of soil organic matter may at least partly replenish the soluble C pool after sampling, even as microbial uptake and respiration deplete it. High respiration rates in the wetter soil samples are likely accompanied by higher rates of enzyme production/diffusion and depolymerization–consequently it is not obvious what the short-term net effect of soil moisture on available C should be.

(2) The CO2 released after wetting of dry soil may come from multiple sources–both endo- and extra-cellular. To the extent that respiration after wetting represents a microbial stress-response or a side effect of microbial stress physiology, the link between available C and respiration is not direct. For instance, if this C represents microbial osmolytes, the size of the pulse might depend more on the propensity of the microbial community to allocate C to osmolytes than C availability persay. Microbes acclimated

to dry soil might accumulate more osmolytes, thus releasing more C after wetting regardless of overall C availability.

(3) Similarly, to the extent that C respired following wetting is derived from extracellular sources, it is unclear whether those sources represent the same C that is readily available under moist conditions versus some more occluded form that is only made available by the physical effects of drying and wetting (see for instance Homyak et al. 2018).

These concepts are really broader critiques of the use of short-term CO2 emissions after wetting as a general metric of soil C availability in the first place. The phenomenon in question is very complex and still not totally understood on a mechanistic level. In the soil-health realm, the relationship between the pulse and C availability is taken as a given. This is appropriate at some level, as it seems plausible that soils that exhibit larger respiration pulses after wetting likely have more microbial biomass, and possibly a more active microbial biomass. However, it would be good to acknowledge that the relationship between C-respired-after-wetting and "available C" (defined as a pool) is not straightforward. I would advocate for a brief but well referenced consideration of the possible mechanisms that might influence the post-wetting respiration pulse: depolymerization, synthesis of osmolytes, and release of occluded C on wetting. Some combination of these mechanisms might explain the findings of this study–but from the perspective of soil health testing the main point is that antecedent soil moisture matters.

Line-by-line comments:

Lines 24-25: This remains an area of active research. Some studies suggest significant microbial mortality on wetting (Blazewicz et al. 2015, 2020); others suggest that the CO2 is derived from osmolytes, but that they might be processed endo-cellularly and that lysis isn't a big player (Slessarev et al. 2020; Warren 2020); yet more studies emphasize the role of wetting in liberating soluble components of (extracellular) soil organic matter (Homyak et al. 2018).

Line 131: In the figure caption, the "standard deviation" referred to here is based on the bootstrap error propagation? Please clarify.

Line 153: "...moisture contents sufficient to oxidize...". Clarify that the microbes do the oxidizing, not the moisture itself.

References:

Blazewicz, S. J., Schwartz, E., & Firestone, M. K. (2014). Growth and death of bacteria and fungi underlie rainfall‐induced carbon dioxide pulses from seasonally dried soil. Ecology, 95(5), 1162-1172.

Blazewicz, S. J., Hungate, B. A., Koch, B. J., Nuccio, E. E., Morrissey, E., Brodie, E. L., ... & Firestone, M. K. (2020). Taxon-specific microbial growth and mortality patterns reveal distinct temporal population responses to rewetting in a California grassland soil. The ISME Journal, 1-13.

Homyak, P. M., Blankinship, J. C., Slessarev, E. W., Schaeffer, S. M., Manzoni, S., & Schimel, J. P. (2018). Effects of altered dry season length and plant inputs on soluble soil carbon. Ecology, 99(10), 2348-2362.

Meisner, A., Leizeaga, A., Rousk, J., & Bååth, E. (2017). Partial drying accelerates bacterial growth recovery to rewetting. Soil Biology and Biochemistry, 112, 269-276.

Meisner, A., Rousk, J., & Bååth, E. (2015). Prolonged drought changes the bacterial growth response to rewetting. Soil Biology and Biochemistry, 88, 314-322.

Miller, A. E., Schimel, J. P., Meixner, T., Sickman, J. O., & Melack, J. M. (2005). Episodic rewetting enhances carbon and nitrogen release from chaparral soils. Soil Biology and Biochemistry, 37(12), 2195-2204.

Slessarev, E. W., Lin, Y., Jiménez, B. Y., Homyak, P. M., Chadwick, O. A., D'Antonio, C. M., & Schimel, J. P. (2020). Cellular and extracellular C contributions to respiration after wetting dry soil. Biogeochemistry, 147(3), 307-324.

Warren, C. R. (2020). Pools and fluxes of osmolytes in moist soil and dry soil that has been re-wet. Soil Biology and Biochemistry, 150, 108012.

---

## Author Comment (AC2) · 24 Nov 2020

1. The basic finding of this paper seems well supported by the data: antecedent moisture history affects the release of $CO_2$ from dry soil following wetting. This observation seems generally consistent with previous studies: for instance, both the duration of drying (Miller et al. 2005; Meisner et al. 2015) and the severity of drying (Meisner et al. 2017) influence respiration after wetting. In this case, soil moisture during the wet period was varied and found to affect the respiration rate after wetting. The novelty of this short note is that it raises this point specifically in the context of soil health testing.

**Response: We appreciate the evaluation that the basic findings are sound and consistent with prior observations.**

2. The specific interpretation advanced in this study–that respiration prior to drying affects the post-wetting respiration pulse specifically by reducing C availability–is only indirectly supported by the data and might need more thought. This interpretation seems to rest on an assumption that there is a fixed pool of available C at sampling, and that any losses of C between sampling and drying/rewetting reduce the size of this pool–resulting in a proportionately smaller pulse. Strictly speaking this is assumption is true of the bulk organic C pool, but it may not apply to the small fraction of that bulk pool that is actually available at any given moment (e.g. the soluble C pool). The apparent balancing of C fluxes observed in this experiment (Fig 3) does seem consistent with the idea of a fixed available C pool–but several factors could make things more complicated:

(1) Depolymerization of soil organic matter may at least partly replenish the soluble C pool after sampling, even as microbial uptake and respiration deplete it. High respiration rates in the wetter soil samples are likely accompanied by higher rates of enzyme production/diffusion and depolymerization–consequently it is not obvious what the short-term net effect of soil moisture on available C should be.

(2) The $CO_2$ released after wetting of dry soil may come from multiple sources–both endo- and extra-cellular. To the extent that respiration after wetting represents a microbial stress-response or a side effect of microbial stress physiology, the link between available C and respiration is not direct. For instance, if this C represents microbial osmolytes, the size of the pulse might depend more on the propensity of the microbial community to allocate C to osmolytes than C availability per se. Microbes acclimated to dry soil might accumulate more osmolytes, thus releasing more C after wetting regardless of overall C availability.

(3) Similarly, to the extent that C respired following wetting is derived from extracellular sources, it is unclear whether those sources represent the same C that is readily available under moist conditions versus some more occluded form that is only made available by the physical effects of drying and wetting (see for instance Homyak et al. 2018).

These concepts are really broader critiques of the use of short-term $CO_2$ emissions after wetting as a general metric of soil C availability in the first place. The phenomenon in question is very complex and still not totally understood on a mechanistic level. In the soil-health realm, the relationship between the pulse and C availability is taken as a given. This is appropriate at some level, as it seems plausible that soils that exhibit larger respiration pulses after wetting likely have more microbial biomass, and possibly a more active microbial biomass. However, it would be good to acknowledge that the relationship between C-respired-after-wetting and "available C" (defined as a pool) is not straightforward. I would advocate for a brief but well referenced consideration of the possible mechanisms that might influence the post-wetting respiration pulse: depolymerization, synthesis of osmolytes, and release of occluded C on wetting. Some combination of these mechanisms might explain the findings of this study–but from the perspective of soil health testing the main point is that antecedent soil moisture matters.

**Response: We agree that our statement that the pulse effect is (by implication exclusively) a C availability response is an over-simplification that deserves an expanded explanation. As helpfully noted, there are a number of potential mechanisms, all difficult to parse in these sorts of assays (and not exclusive to this study). These are excellent comments. Indeed, we have taken the simple soil health path in our discussion of effects, and agree that acknowledging alternatives would be helpful and appropriate. To that end, we propose:**

a) **In the Introduction, note why this index is often used as an indicator of C availability for soil health tests. In particular, we will note Franzleubbers (2000) wherein assay responses were correlated with microbial biomass carbon, soil organic carbon, and particulate organic carbon. We will also note the potential mechanisms behind the Birch effect (Jarvis et al. 2007).**

b) **In the Discussion, we will make it clear that the correction factor is not intended to imply that there is a fixed C pool, rather to demonstrate the way antecedent conditions could affect soil health tests and offer some potential solutions. We will therefore adapt our language to represent this method from a soil health perspective rather than as a direct indication of available C, acknowledging as suggested that the relationship between C-respired-after-wetting and the available C pool is not straightforward.**

c) **Additionally we will discuss the potential effects of depolymerization, microbial stress, and extracellular C sources along with other Birch effect mechanisms to point out that while they are unlikely responsible for the results we observed (i.e., more loss during dry-down results in a lower $CO_2$ pulse after re-wetting), that using a correction factor may be complicated by them. For example, as referenced, depolymerization rates increase with higher levels of soil moisture, in turn resulting in higher rates of C replenishment in wetter soils (Wild, et al., 2014). Therefore, we would expect wetter soils with higher depolymerization rates to result in higher flushes of C, which is not consistent with our finding that wetter soils had a smaller $CO_2$ flush upon re-wetting (Fig. 2), a result most likely due to $CO_2$ loss during drying (Fig. 1b).**

3. Lines 24-25: This remains an area of active research. Some studies suggest significant microbial mortality on wetting (Blazewicz et al. 2015, 2020); others suggest that the CO2 is derived from osmolytes, but that they might be processed endo-cellularly and that lysis isn't a big player (Slessarev et al. 2020; Warren 2020); yet more studies emphasize the role of wetting in liberating soluble components of (extracellular) soil organic matter (Homyak et al. 2018).

**Response: Agreed, and suggested references are duly noted and helpful. We will incorporate them throughout the Introduction and Discussion.**

4. Line 131: In the figure caption, the "standard deviation" referred to here is based on the bootstrap error propagation? Please clarify.

**Response: Yes, standard deviation is based on bootstrapping the total amount of $CO_2$ loss during dry-down 10,000 x. We will clarify this in the figure caption.**

5. Line 153: ". . .moisture contents sufficient to oxidize. . .". Clarify that the microbes do the oxidizing, not the moisture itself

**Response: Agreed, we will clarify.**

**References**

**Franzluebbers, A. J., Haney, R. L., Honeycutt, C. W., Schomberg, H. H., & Hons, F. M. (2000). Flush of carbon dioxide following rewetting of dried soil relates to active organic pools. *Soil Science Society of America Journal*, *64*(2), 613-623.**

**Jarvis, P., Rey, A., Petsikos, C., Wingate, L., Rayment, M., Pereira, J., ... & Manca, G. (2007). Drying and wetting of Mediterranean soils stimulates decomposition and carbon dioxide emission: the "Birch effect". *Tree physiology*, *27*(7), 929-940.**

**Wild, B., Ambus, P., Reinsch, S., & Richter, A. (2018). Resistance of soil protein depolymerization rates to eight years of elevated CO 2, warming, and summer drought in a temperate heathland. *Biogeochemistry*, *140*(3), 255-267.**

---

## Author Response (AR1)

**Topical Editor comments**

Dear Authors,

I have read both the reviewer reports and your manuscript carefully, and recommend that some minor revisions be undertaken according to the reviewers' comments. Thank you for your thoughtful responses to those comments, which I feel capture well the revisions that need to be done (especially the response to Reviewer #2). With regard to the reviewer suggestions of putting this work in context of previous work on soil drying-rewetting and $CO_2$ efflux from soils, I recommend including this to the extent possible given the format of the short communication.

**Response: Thank you for the clear directions on how to proceed with the revision. We revised the manuscript in response to Reviewer #2's major comments, while briefly putting this work into context as suggested by Reviewer #1. Additionally, we made the line-by-line revisions suggested by both reviewers. More details are below.**

**Reviewer #1 comments**

1. There is considerable research on soil moisture content and biological activity of soils, including dry-down and wet-up. This should be discussed more in Introduction and Discussion.

**Response: For the sake of brevity in this "short communication", we did not review the many studies on soil moisture content and $CO_2$ pulses. However, we have included a more nuanced discussion of the Birch effect in the Introduction (see lines 30 – 36).**

There are several studies that show that the drier soils are in the field, they tend to release more CO2 upon re-wetting regardless of dry-down C loss. Thus, there are biophysical mechanisms at play other than dry-down C loss.

**Response: In response to your comment, we specifically addressed why these mechanisms are unlikely to be responsible for our results in the Discussion (see lines 166 – 171).**

Figure 3 still shows more CO2-C from the 30.

**Response: Differences in total C loss among treatments were not significant (one-way ANOVA: P = 0.28). More importantly, the proportion of pre-assay loss was substantially smaller (Fig. 3).**

2. These findings need to be placed in context of the purpose and practicalities of CO2 Burst test. Drying is often needed, if not necessary, to stabilize microbial activity before analyzing for CO2 production. Whereas accounting for C loss during drying might not be feasible or possible for commercial and even research labs. This would be very onerous. How much more are we gaining by accounting for this C?

**Response: We agree that our findings are inconvenient for an assay that requires drying and that calculating a correction factor is an added complexity. As we point out in the discussion (lines 173 – 177 & 187 – 191), a correction factor is not needed if comparing soils of similar moisture contents at collection or perhaps if the dry-down period is fast. However, not correcting for soils at different moistures will likely lead to erroneous conclusions about carbon availability.**

This paper has not convinced me that we gain much. Despite finding differences between 30, this paper would be much stronger if it included a comparison of two or more treatments and showed that measuring CO2 Burst at different moisture contents obscured our ability to detect treatment differences.

**Response: We agree that pre-assay dry-down effects will likely differ not just by treatment but also by soil type and structure; this is why we discuss the need for calculating correction factors on a case-by-case basis (lines 183 – 185). Little would be gained by adding different treatments to this study because we fully expect correction factors to differ.**

There are always experimental artifacts with incubation-based, laboratory measurements of soils. A few studies have shown that using more soil reduces variability. The most important thing is that we treat soils the same across time or space, and that the methodology is not creating confounding effects.

**Response: We agree that it is important that methodology not create confounding effects, which is why we chose to conduct this study. Our results reveal that initial soil moisture can confound our interpretation of carbon availability from the $CO_2$ pulse assay. We want others to be aware of this limitation, and so we provide a correction and a few alternatives for minimizing $CO_2$ losses during dry-down.**

SPECIFIC COMMENTS:

L24. Replace 'the reintroduction of moisture' with 'rewetting'

**Response: We have made this change (see line 27).**

L33. Delete 'different'

**Response: We have made this change (see line 45).**

L38. How was it collected? Shovel, or soil probe? More details are needed here.

**Response: We collected the soil using a shovel and have added this to the manuscript (see line 50).**

L43. So total MAP is 2305? This seems very high.

**Response: No, snowfall has an average density of ~10% so 1300 mm represents only ~130 mm of precipitation. Nevertheless "average annual rainfall" should read "average annual precipitation" to include snowfall. Thank you for identifying this issue. We changed the sentence to make it clear that MAP is 1005 mm (not 2305 mm) and that the site receives 1.3 m of snowfall on average (see lines 55 – 56).**

L49. What was the initial water content of the field soil when you collected it? Was it below 30

**Response: Yes, the initial soil moisture was 11% WFPS (see lines 69 – 70).**

L85. This seems overly complicated. Why not use area-under-the-curve to calculate cumulative CO2?

**Response: We did in fact calculate the area under the curve, using bootstrapping to define the curve to allow for error estimates. We edited the text for clarification (see lines 103 – 104).**

L110. Why does respiration go back up at 8d? This is interesting and looks like there might be treatment effect?

**Response: The slight increase in $CO_2$ respiration at Day 8 may be due to more humid incubation conditions later in the experiment as evidenced by the slight increase in soil moisture (Fig. 1a).**

L131. Why use standard deviation in this graph? Fig. 2 uses standard error. I suggest being consistent. Also, use same colors in Fig. 1 and 2 for consistency. Place letters to abbreviate significant differences among means in both Fig. 1 and 2.

**Response: SD is more appropriate than SE for Fig 3, which reports error from the results of bootstrapping (see Fig. 3 legend and lines 106 – 114). Thank you for the suggestions about consistent colors and letters. We have updated Fig. 3 to reflect this suggestion (see line 145).**

1. The basic finding of this paper seems well supported by the data: antecedent moisture history affects the release of $CO_2$ from dry soil following wetting. This observation seems generally consistent with previous studies: for instance, both the duration of drying (Miller et al. 2005; Meisner et al. 2015) and the severity of drying (Meisner et al. 2017) influence respiration after wetting. In this case, soil moisture during the wet period was varied and found to affect the respiration rate after wetting. The novelty of this short note is that it raises this point specifically in the context of soil health testing.

**Response: We appreciate the evaluation that the basic findings are sound and consistent with prior observations.**

2. The specific interpretation advanced in this study–that respiration prior to drying affects the post-wetting respiration pulse specifically by reducing C availability–is only indirectly supported by the data and might need more thought. This interpretation seems to rest on an assumption that there is a fixed pool of available C at sampling, and that any losses of C between sampling and drying/rewetting reduce the size of this pool–resulting in a proportionately smaller pulse. Strictly speaking this is assumption is true of the bulk organic C pool, but it may not apply to the small fraction of that bulk pool that is actually available at any given moment (e.g. the soluble C pool). The apparent balancing of C fluxes observed in this experiment (Fig 3) does seem consistent with the idea of a fixed available C pool–but several factors could make things more complicated:

(1) Depolymerization of soil organic matter may at least partly replenish the soluble C pool after sampling, even as microbial uptake and respiration deplete it. High respiration rates in the wetter soil samples are likely accompanied by higher rates of enzyme production/diffusion and depolymerization–consequently it is not obvious what the short-term net effect of soil moisture on available C should be.

(2) The $CO_2$ released after wetting of dry soil may come from multiple sources–both endo- and extra-cellular. To the extent that respiration after wetting represents a microbial stress-response or a side effect of microbial stress physiology, the link between available C and respiration is not direct. For instance, if this C represents microbial osmolytes, the size of the pulse might depend more on the propensity of the microbial community to allocate C to osmolytes than C availability per se. Microbes acclimated to dry soil might accumulate more osmolytes, thus releasing more C after wetting regardless
of overall C availability.

(3) Similarly, to the extent that C respired following wetting is derived from extracellular sources, it is unclear whether those sources represent the same C that is readily available under moist conditions versus some more occluded form that is only made available by the physical effects of drying and wetting (see for instance Homyak et al. 2018).

These concepts are really broader critiques of the use of short-term $CO_2$ emissions after wetting as a general metric of soil C availability in the first place. The phenomenon in question is very complex and still not totally understood on a mechanistic level. In the soil-health realm, the relationship between the pulse and C availability is taken as a given. This is appropriate at some level, as it seems plausible that soils that exhibit larger respiration pulses after wetting likely have more microbial biomass, and possibly
a more active microbial biomass. However, it would be good to acknowledge that the relationship between C-respired-after-wetting and "available C" (defined as a pool) is not straightforward. I would advocate for a brief but well referenced consideration of the possible mechanisms that might influence the post-wetting respiration pulse: depolymerization, synthesis of osmolytes, and release of occluded C on wetting. Some combination of these mechanisms might explain the findings of this study–but from the perspective of soil health testing the main point is that antecedent soil moisture matters.

**Response: We agree that our statement that the pulse effect is (by implication exclusively) a C availability response is an over-simplification that deserves an expanded explanation. As helpfully noted, there are a number of potential mechanisms, all difficult to parse in these sorts of assays (and not exclusive to this study). These are excellent comments. To that end, we revised the manuscript as follows:**

a) **In the Introduction, we noted why this index is often used as an indicator of C availability for soil health tests (see lines 21 – 24). We also included the potential mechanisms behind the Birch effect as reviewed by Jarvis et al., 2007 (see lines 30 – 36).**

b) **In the Discussion, we clarified that the correction factor is not intended to imply that there is a fixed C pool, rather to demonstrate the way antecedent conditions could affect soil health tests and offer some potential solutions (see lines 181 – 183). We adapted our language to represent this method from a soil health perspective rather than as a direct indication of available C (see throughout entire manuscript).**

c) **Additionally, we discussed the potential Birch effect mechanisms in the introduction (see lines 30 – 36). We then pointed out in the discussion why they are unlikely responsible for the results we observed (i.e., more loss during dry-down resulting in a lower $CO_2$ pulse after re-wetting; see lines 166 – 171).**

3. Lines 24-25: This remains an area of active research. Some studies suggest significant microbial mortality on wetting (Blazewicz et al. 2015, 2020); others suggest that the CO2 is derived from osmolytes, but that they might be processed endo-cellularly and that lysis isn't a big player (Slessarev et al. 2020; Warren 2020); yet more studies emphasize the role of wetting in liberating soluble components of (extracellular) soil organic matter (Homyak et al. 2018).

**Response: Agreed, and suggested references are duly noted and helpful. We incorporated them into the introduction (please see lines 31 – 36).**

4. Line 131: In the figure caption, the "standard deviation" referred to here is based on the bootstrap error propagation? Please clarify.

**Response: Yes, standard deviation is based on bootstrapping the total amount of $CO_2$ loss during dry-down 10,000 x. We clarified this in the figure caption (see lines 150 – 151).**

5. Line 153: ". . .moisture contents sufficient to oxidize. . .". Clarify that the microbes do the oxidizing, not the moisture itself

**Response: We clarified this in line 183.**

---

## Author Response (AR2)

**Dear Dr. Lavallee,**

**Thank you for the attention to detail and the revisions you suggested that improve the clarity of our manuscript. Below is our point-by-point response.**

Line 39: perhaps add "and different timepoints within the same soil" or similar, but you may choose to ignore this comment.

**Response: Agreed (see Lines 39-40)**

Line 51: I believe this is still slightly confusing, since annual precipitation includes snowfall and so the reader may still expect it to be a larger number than the annual snowfall. It's not clear whether the snowfall number presented is snow depth or meltwater. Perhaps simply report total annual precipitation, which need not be divided into rainfall and snowfall for the purposes of this manuscript

**Response: Agreed, we removed the snowfall information (see Line 52)**

.Lines 52-53: change "for the prior 40 years" to " for the 40 years prior to sampling"

**Response: Done (see Lines 53-54)**

Line 63: change "previously assessed (Robertson, 2016)" to "as previously assessed by Robertson (2016)"

**Response: Done (see Line 64)**

Line 98: Consider changing "using bootstrapping to allow for error estimates" to "using bootstrapping to estimate error"

**Response: Done (see Lines 98-99)**

Line 101: "zeroes" should be "zeros"

**Response: Done (see Line 103)**

Line 143-144: This feels too strong. Perhaps "thus calling into question its reliability as a soil health indicator"?

**Response: Agreed (see Lines 145-146)**

Line 145: add "subsequent" before "short-term CO2 pulses"

**Response: We have deleted this sentence based on the recommendation in the following comment (see Lines 147-148)**

Lines 145-146: I believe that "thereby diminishing the assay's value as an indicator of soil health" is a bit too strong and is also repetitive with the first sentence in the paragraph. I recommend deleting, and this paragraph could be combined with the following one.

**Response: Agreed (see Lines 147-148)**

Line 148: Consider changing to "Without knowledge of these pre-assay CO2 losses..."

**Response: Done (see Lines 148-149)**

Line 152: change "daily" to 24-h

**Response: Done (see Line 153)**

Lines 155-160: If brevity is possible here, it would be helpful to the reader if you could clarify these mechanisms in the context of the experimental stages, i.e. would these potential mechanisms occur in or affect results of the dry-down phase or the rewetting and 24-h pulse?

**Response: We have clarified that (see Line 157 and 159).**

Line 171: change to "fixed, available C pool"

**Response: Done (see Line 171)**

Line 171-172: Please add a bit of detail as to why soil-specific correction factors are needed, e.g. differences in soil texture or microbial communities.

**Response: C quality and microbial communities vary with soil type and seasonal variation, both of which are potential reasons for soil-specific correction factors as detailed in Lines 173-174**

Line 183: change to "methods and/or a correction factor"

**Response: Done (see Line 185)**

**Respectfully,**

**Matt Belanger, Carmella Vizza, Phil Robertson, and Sarah Roley**

---

## Author Response (AR3)

Thank you, Dr. Lavallee for your helpful comments throughout the editorial process. We truly value your feedback and enjoyed working with you.

Sincerely,

Carmella Vizza on behalf of all authors